# The effect of an additional pre-extubational loading dose of caffeine citrate on mechanically ventilated preterm infants (NEOKOFF trial): Study protocol for a multicenter randomized clinical trial

Kinga Kovács[1,2,3], Rita Nagy[1,4], Lilla Andréka[5], Brigitta Teutsch[1,6,7], Miklós Szabó[2,3], Péter Varga[2,3], Péter Hegyi[1,7,8], Péter Hársfalvi[1,9], Nándor Ács[1,2], Ágnes Harmath[2,3], Csaba Nádor[2,3], Ákos Gasparics[1,2,3]*

1 Centre for Translational Medicine, Semmelweis University, Budapest, Hungary, 2 Department of Obstetrics and Gynecology, Semmelweis University, Budapest, Hungary, 3 Department of Neonatology, Semmelweis University, Budapest, Hungary, 4 Heim Pál National Pediatric Institute, Budapest, Hungary, 5 Faculty of Medicine, Semmelweis University, Budapest, Hungary, 6 Department of Radiology, Medical Imaging Centre, Semmelweis University, Budapest, Hungary, 7 Institute for Translational Medicine, University of Pécs, Medical School, Pécs, Hungary, 8 Institute of Pancreatic Diseases, Semmelweis University, Budapest, Hungary, 9 Department of Biostatistics, University of Veterinary Medicine, Budapest, Hungary

☯ These authors contributed equally to this work.
* gasparics.akos@semmelweis.hu

**Data Availability Statement:** No datasets were generated or analysed during the current study. All

## Abstract

### Background

Minimizing the duration of mechanical ventilation is one of the most important therapeutic goals during the care of preterm infants at neonatal intensive care units (NICUs). The rate of extubation failure among preterm infants is between 16% and 40% worldwide. Numerous studies have been conducted on the assessment of extubation suitability, the optimal choice of respiratory support around extubation, and the effectiveness of medical interventions. Since the Caffeine Therapy for Apnea of Prematurity (CAP) trial, caffeine has become one of the essential drugs at NICUs. However, the optimal dosage and timing for adequate effectiveness still need to be more conclusive. Previous studies suggest that higher doses of caffeine treatment increase the success rate of extubation. Therefore, we aim to determine whether using a single additional loading dose of caffeine citrate one hour prior to extubation impacts the success rate of extubation.

### Methods

The study is an open-label, multicenter randomized clinical trial testing the effectiveness and safety of pre-extubational loading dose of caffeine citrate. Inclusion criteria will be infants born before the 32nd gestational week, before the first extubation attempt after at least 48 hours of mechanical ventilation, and a signed parental informed consent. A total of 226 patients will be randomly allocated to either the experimental or control group. The

relevant data from this study will be made available upon study completion.

**Funding:** The study will be financially sponsored by the Department of Obstetrics and Gynecology, Semmelweis University, Budapest. Additional funding for the study has been provided by the grant TINL 3.2.1-21-2022-00011.

**Competing interests:** The authors have declared that no competing interests exist.

randomization will be stratified by gestational age and antenatal steroid prophylaxis. Preterm infants in the experimental group will receive an additional intravenous (IV) loading dose (20 mg/kg) of caffeine citrate one hour before the first planned extubation, in addition to the standard dosing regimen (20 mg/kg caffeine citrate IV on the first day of life and 5 to 10 mg/kg IV or orally caffeine citrate each consecutive day). Preterm infants in the control group will receive the standard dosing regimen. The primary outcome will be reintubation within 48 hours.

## Discussion

A pre-extubational loading dose of caffeine citrate can reduce extubation failure. Obtaining evidence on this feature has the potential to contribute to finding the optimal dosing regimen.

## Trial registration number

The study protocol was approved by the Hungarian Ethics Committee for Clinical Pharmacology of the Medical Research Council and National Institute of Pharmacy and Nutrition (OGYÉI/6838-11/2023). ClinicalTrials.gov identifier NCT06401083 Registered 06. May 2024.; EudraCT number: 2022-003202-77.

## Introduction

### Background and rationale

In recent decades, the survival rate of preterm infants has increased significantly [1, 2]. According to the 2017 data of the Hungarian Statistical Office (KSH), the mortality rate of preterm infants born in Hungary with a birth weight below 1000 grams has decreased by a quarter since 1990 [3]. One of the crucial goals of neonatology, in addition to improving survival rates, is to reduce the rate of complications in surviving preterm infants [4]. Adverse effects on the immature lungs of preterm infants, such as intrauterine inflammation, oxygen toxicity, baro-, and volutrauma caused by mechanical ventilation, lead to one of the most common complications of preterm birth, bronchopulmonary dysplasia (BPD). Preterm infants with BPD have a higher incidence of childhood asthma-like symptoms and upper respiratory tract diseases. Respiratory function studies in childhood have shown that preterm infants with BPD have lower lung function than their full-term peers [5].

Non-invasive ventilatory support techniques have been promoted in neonatology to reduce the rate of BPD. Nevertheless, a proportion of preterm infants still require mechanical ventilation during their care. According to the Hungarian National Neonatal Database, in 2021, around 34% of preterm neonates born before the 32nd gestational week required mechanical ventilation for more than 24 hours in 2021. The more immature the preterm infants are, the higher this ratio will be.

One of the most essential therapeutic objectives in the management of preterm infants requiring invasive ventilatory support is to minimize the time required for mechanical ventilation [6] in order to reduce the rate of BPD [7] and to improve neurodevelopmental outcomes [8, 9]. Although the aim is to extubate as early as possible, there are no unanimous recommendations on timing and conditions. Among preterm infants, the incidence of extubation failure,

i.e. reintubation within the next 48–72 hours after changing to non-invasive ventilation methods, ranges between 16% and 40%. Extubation failure is associated with more intensive respiratory support than before extubation, airway injury, and hemodynamic instability [10]. Optimizing the time and conditions of extubation are essential, as reintubation and repeated mechanical ventilation are accompanied by additional stress and, therefore, potential additional morbidities in preterm infants [11].

There are strategies to optimize the success of extubation in preterm infants: weaning protocols, target saturation, PaCO2 levels, or volume-targeted ventilation [12]. The decision to extubate is currently at the discretion of the attending physician, as there are no objective estimators and standardized guidelines. Several factors can be taken into account as part of the assessment of extubation readiness, such as gestational age and chronologic age at extubation, blood gas pH and inhaled FiO2 before extubation [13], hemoglobin concentration, or the presence of early-onset sepsis [14], and respiratory severity score in the first 6 hours of life [15]. There are rather inconclusive reports on spontaneous breathing trials [16].

After extubation, the use of continuous positive airway pressure (CPAP) or nasal intermittent positive pressure ventilation (NIPPV) is recommended immediately after extubation with a positive end-expiratory pressure (PEEP) higher than 5 $H_2O$cm [12].

A common cause of failure of non-invasive ventilatory support is poor spontaneous respiratory activity in preterm infants and recurrent respiratory arrest (apnea) [17]. Apnea is defined as a respiratory failure of 15–20 seconds or less, associated with bradycardia or desaturation. Apnea develops in preterm infants due to prematurity of the respiratory center, chemoreceptors, and reduced upper airway patency [18].

The apnea-reducing effects of respiratory center stimulant methylxanthines have been known for more than 40 years [19]. On the basis of current knowledge, caffeine is the drug of choice for the medical treatment of apnea. Among methylxanthines, caffeine has the narrowest spectrum of side effects, the broadest therapeutic range, and the longest half-life [9]. In the Caffeine for Apnea of Prematurity (CAP) prospective, randomized clinical trial of 2000 preterm infants, caffeine reduced the development of BPD [20], improved survival without developmental neurological impairment [21], and promoted weaning from mechanical ventilation [18].

Since the CAP study, caffeine use in preterm infants has become common worldwide. Caffeine is currently one of the most commonly administered drugs in neonatal intensive care units (NICUs) [19]. The standard dosing recommendation is a maintenance dose of 5–10 mg·$kg^{-1}$ per day after a loading dose of 20 mg·$kg^{-1}$ of caffeine citrate [22]. Higher loading and maintenance doses have also been investigated in some studies, with some reports suggesting that higher doses of caffeine increase the chance of successful extubation [23–27]. However, other studies have reported more frequent adverse effects at higher doses, such as tachycardia or cerebellar injury [28, 29]. A recent Cochrane review found that higher doses did not increase mortality and were not associated with adverse effects. High-dose caffeine also has the potential to reduce the rate of BPD. There is a lack of studies regarding long-term neurodevelopmental outcomes [30].

Due to the somewhat inconclusive literature, caffeine dosing varies between institutions [31]. Further clinical studies are needed to determine the optimal dose [32].

## Objectives

We seek to answer the question whether using a single additional loading dose of caffeine citrate one hour prior to extubation affects the success rate of extubation. In addition, we would like to assess the frequency and severity of side effects.

## Methods

### Trial design

To investigate the effect of a pre-extubational loading dose of caffeine citrate, we plan to carry out a two-armed randomized open-label multicenter clinical trial, including preterm infants treated in a tertiary intensive care unit at Semmelweis University (Department of Obstetrics and Gynaecology, Baross Street division; Department of Obstetrics and Gynaecology, Üllői Street division; and Pediatric Center, Bókay Street Department). A total of 226 patients are planned to be enrolled. In our units, according to the institutional protocol, preterm infants before week 32 of gestation receive standard dosed caffeine therapy. This covers a maintenance dose of 5 mg/kg of caffeine citrate administered intravenously (iv) once or twice daily after a loading dose of 20 mg/kg on the first day of life.

Preterm infants who have been on mechanical ventilation for at least 48 hours before attempted extubation are planned to be randomly allocated into intervention (113 participants) and control (113 participants) groups. Stratification will be based on gestational age and steroid prophylaxis. The intervention is an additional loading dose (20 mg/kg) of caffeine citrate IV 60 minutes before extubation. The control group will receive the above-mentioned routine dosing regimens. The trial has been designed according to the Standard Protocol Items: Recommendations for Intervention Trials -SPIRIT guidelines [33] (Figs 1 and 2 and S1 File)

The target population will be preterm infants treated in a tertiary intensive care unit at Semmelweis University regardless of chronological age. Eligibility will be assessed based on exclusion and inclusion criteria.

Inclusion criteria are as follows:

- Preterm infants born before 32nd week of gestation is completed;

- Were mechanically ventilated for at least 48 hours;

- Before the first planned extubation;

- Received surfactant treatment.

  Exclusion criteria are:

- Lack of informed consent, refusal to participate in the study by parents or legal guardians;

- Major congenital anomaly;

- Hydrops foetalis;

- Persistent tachycardia before extubation, fetal/neonatal arrhythmia;

- Asphyxia;

- Non-invasive ventilation was attempted at prior unplanned extubation (Fig 1).

A trained clinician will explain the study design and methodology in detail to parents and take informed consent during mechanical ventilation. They will clarify and define the potential risks and benefits and answer all the questions the parents have (S2 File). If any of the criteria are not met at the time of planned extubation, or an unplanned extubation occurred and non-invasive ventilation was attempted before re-intubation, preterm infants will be considered as not eligible for the study.

No biological specimens will be collected.

| | STUDY PERIOD | | | | | |
|---|---|---|---|---|---|---|
| | Enrolment | Allocation | Post-allocation | | | Close-out |
| TIMEPOINT* | $-t_1$ | 0 | $t_1$ | $t_2$ | $t_3$ | $t_4$ |
| **ENROLMENT:** | | | | | | |
| **Eligibility screen** | X | | | | | |
| **Informed consent** | X | | | | | |
| **Allocation** | | X | | | | |
| **INTERVENTIONS:** | | | | | | |
| *Intervention arm A* (Additional loading dose [20 mg/kg] of caffeine citrate 1 hour before extubation) | | | X | | | |
| *No intervention arm B* (Standard dosing of caffeine citrate before extubation) | | | X | | | |
| **ASSESSMENTS:** | | | | | | |
| *Complete A form* | X | X | | | | |
| *Primary outcome assessment* (Reintubation within the next 48 hours after extubation) | | | | X | | |
| *Complete B form* | | | | X | | |
| *Complete C form* | | | | | X | |
| *Complete D Form* | | | | | | X |

** $t_{-1}$: after intubation of a preterm neonate who met the eligibility criteria, 0: 1-2 hours before extubation, $t_1$: extubation $t_2$: 48 hours after extubation; $t_3$: at discharge from the participating unit, $t_4$: At two years of corrected age.

**Fig 1. Schedule of enrollment, interventions and assessments according to the standard protocol items: Recommendations for interventional Trials 2013 statement.**

## Intervention

It is designed to be a two-armed study, with the intervention of an additional loading dose of caffeine citrate (CITRATE DE CAFEINE COOPER; Coopération Pharmaceutique Française, Melun, France, S3 File) 60 minutes before planned extubation. Sixty minutes post-administration is considered ideal for extubation due to the 20-minute-long infusion and the increasing contractility of the diaphragm, which reaches its peak 25–30 minutes after administration [34].

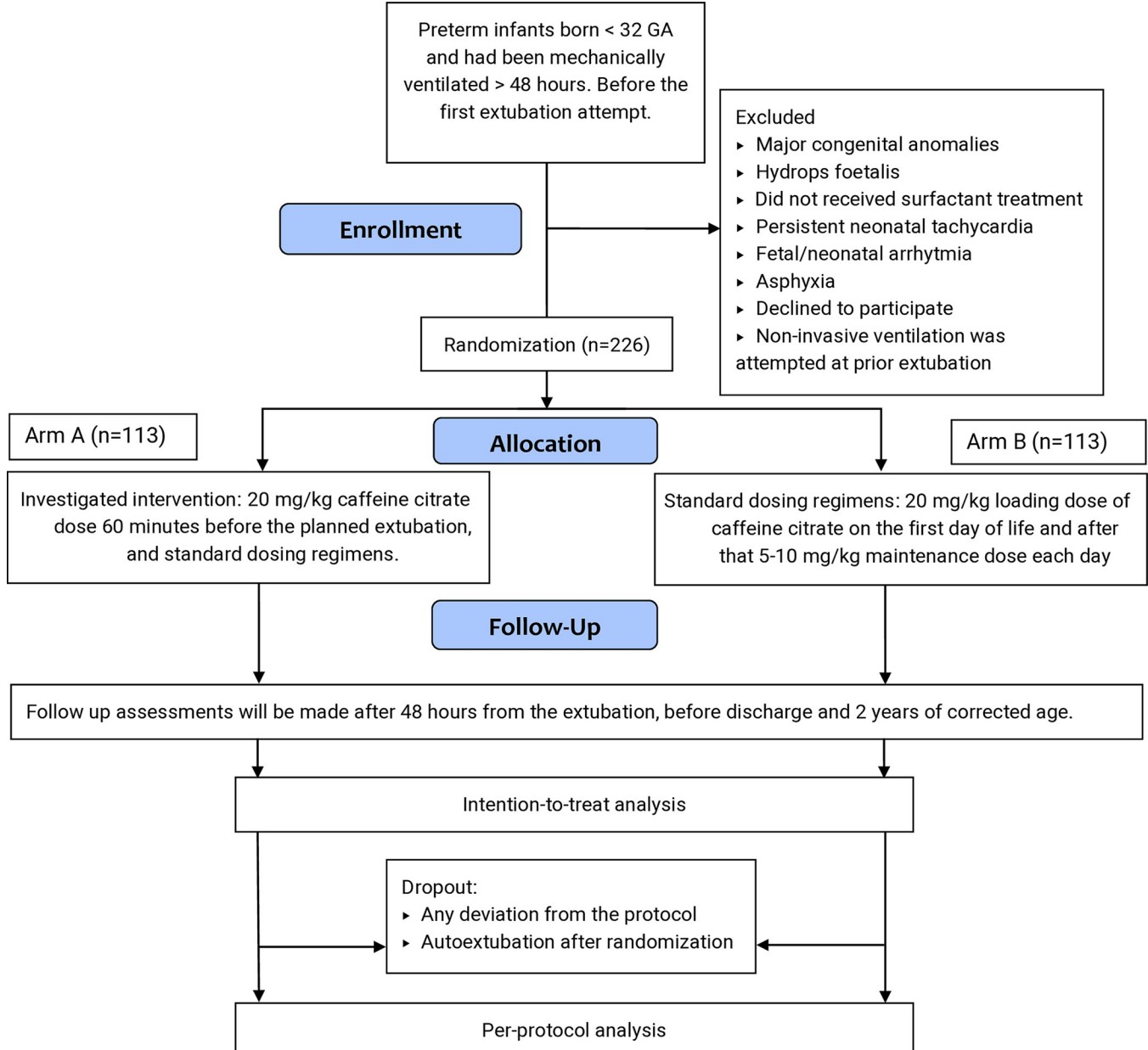

**Fig 2. SPIRIT flow chart.** SPIRIT, Standard Protocol Items: Recommendations for Interventional Trials, GA, Gestational Age, Consent and participants.

Arm "A" (intervention group) will receive iv. 20 mg/kg loading dose of caffeine citrate IV on the first day of life, then 5mg/kg maintenance dose IV once or twice each day. Participants in this arm will receive a 20 mg/kg caffeine citrate IV dose again before the planned extubation. In the intervention arm, under 1000 grams of birth weight, the caffeine citrate will be diluted to 1 ml, and the infusion rate will be 3 mL/hr. Between 1000–2000 grams, caffeine citrate will be diluted to 2 mL and administered at an infusion rate of 6 mL/hr. Above 2000 grams, caffeine citrate is diluted to 3 mL with an infusion rate of 9 mL/hr. Caffeine citrate will be diluted with a maintenance infusion; in the case of total enteral feeding, it will be diluted with a 5% dextrose solution.

Arm "B" (control group) will receive a 20 mg/kg loading dose of caffeine citrate IV on the first day of life, then 5–10 mg/kg maintenance dose IV each day, also on the day of the extubation.

Any deviation from the protocol (e.g., administering the subsequent dose of caffeine citrate earlier or using higher doses) will be recorded and reported by the Steering Committee (SC) to the Data Monitoring Committee and will be treated as a dropout.

Extubation will take place when a preterm infant is considered ready, at the discretion of a senior clinician. After extubation, NIPPV or synchronized NIPPV CPAP (according to local protocol) will be implemented immediately with a minimum of 7 $H_2Ocm$ PEEP.

## Sample size

Sample size considerations and primary efficacy analysis: The sample size of the study was calculated based on the expected extubation failure rate.

We hypothesize that the reintubation rate in the arm receving the loading dose (Arm "A") is expected to be 20%, calculated from cases at our unit where the pre-extubational loading dose of caffeine citrate was applied. The expected reintubation rate of the arm that did not receive the loading dose (Arm "B") is 36.8% [27]. The calculated sample size is 226 patients (113 participants/arm). On the basis of our results, this sample size would provide 80% statistical power to detect a significant difference in chi-square using a 5% significance level test. The drop-out rate is expected to be minimal due to the specific population and the short timeframe between the enrollment, randomization, and the one-time intervention.

## Statistical analyses

We will conduct both a per-protocol analysis, which focuses on participants who completed the study according to protocol requirements, and an intention-to-treat analysis, encompassing all patients who received the intervention except for study withdrawals, for all outcomes. In our final analysis, we will prioritize intention-to-treat analysis over per-protocol analysis. We expect no missing values for the primary endpoint.

Descriptive statistics, including both counts and percentages, will summarize baseline characteristics of patients. Changes from baseline to follow-up periods and end-of-trial visits for continuous variables will be presented using central tendency and dispersion measures. Dichotomous variables will be reported with absolute and relative frequencies. Continuous variables will be compared using the t-test or Mann-Whitney test, whereas the $\chi^2$ test or Fisher's exact test will be used for dichotomous outcomes [35]. Results will be deemed statistically significant if the p-value is <0.05. Statistical analysis will be performed using the R software package.

## Random allocation, data collection

Randomization of preterm infants will take place after informed consent of parents and eligibility assessment. The allocation ratio will be 1:1. Stratification will be based on gestational age and antenatal steroid prophylaxis. The randomization sequence will be generated using an algorithm called Big-Stick Design, using the R programming language. Random allocation and data collection will be performed via the REDCap software (Research Electronic Data Capture).

## Blinding

Due to the nature of the study population (age and status of the neonates), participants do not need to be blinded and neither those performing the intervention nor treating clinicians will

be blinded. To reduce bias, the main statistical analyses to evaluate the hypotheses will be performed by a statistician masked to the allocation of participants.

## Outcomes

**Primary outcome.** Our primary endpoint will be the need for reintubation within the next 48 hours after extubation. Extubation failure is usually defined as a time frame of 48 or 72 hours [36]. We chose 48 hours for our primary endpoint because our observations at our units and the literature suggest that approximately 70% of reintubations occur within the first 48 hours after extubation [37]. Furthermore, caffeine half-life ranges from 20 to 100 hours, so during this period we expect significantly higher caffeine levels in the intervention group with the additional loading dose of caffeine-citrate compared to control group with standard care [38]. As far as we know, there are still no standardized, evidence-based guidelines for intubation or reintubation [39], so reintubation will be performed based on the judgment of a senior clinician.

**Secondary outcomes.** A large number of parameters will be monitored during the study. Form A will contain perinatal characteristics and baseline parameters (S4 File). Meanwhile, Form B will assess the first 48 hours after extubation (S5 File). Within Form B, we will assess the number of apneas and expected adverse effects such as tachycardia, elevated blood pressure, or reduced gastric emptying (by measuring gastric residuals). Form C (S6 File) will contain parameters assessed before discharge and compared to the pre-extubational data (e.g., progression or development of intraventricular hemorrhage, and periventricular leukomalacia or necrotizing enterocolitis). BPD will be evaluated at week 36 of postmenstrual age [40]. In case of transfer to another NICU before this date, we will collect these data from admitting institutions. Additionally, we are planning to assess the neurodevelopmental outcome at two years of corrected age, see Form D (S7 File). Appointments will be recorded on the discharge paper for the first follow-up meetings. See Fig 1 for the schedule of assessments.

## Data handling

Data will be managed by the REDCap software. Electronic Case Report Forms (eCRFs) will be used, and the principal investigator will ensure that the data in the eCRF are accurate and complete. Missing or implausible data will be referred to the Chief Investigator using upper and lower limits for continuous variables. Identification numbers and personal details of participants will only be accessible to those directly involved in the research, and stored separately and securely from other data. Members of the Data Monitoring Committee will be delegated investigators, biostatisticians, and data managers.

## Interim analyses and premature termination of the study

Regular assessments will be used to stop the trial prematurely if a study group shows a significant benefit or harm. An interim analysis will be performed when 50% of patients have been enrolled and discharged from hospital.

## Centers

The clinical trial will start in tertiary neonatal intensive care units at Semmelweis University, and will be opened in other centers that meet the compulsory requirements. These requirements are as follows: (1) it needs to be a tertiary neonatal intensive care unit; (2) the center must designate a clinician and a nurse/administrator responsible for adherence to the local study protocol; and (3) the entire team must attend a preliminary meeting where all details

concerning the studies are discussed thoroughly. Centers that wish to participate in the trial need to send a letter of intent by e-mail to the corresponding author.

## Monitoring and safety

Data and implementation monitoring will be performed by the SC. All adverse events and any protocol deviation will be reported to the SC, and after confirmation, these will be reported to the Data Monitoring Committee and the relevant institutional ethics committee (Hungarian Ethics Committee for Clinical Pharmacology of the Medical Research Council and National Institute of Pharmacy and Nutrition).

The SC will be led by ÁG (chief investigator, pediatric specialist, neonatologist). The members will be KK (principal investigator, medical doctor, PhD student), MSz (pediatric specialist, neonatologist), ÁH (pediatric specialist, neonatologist), CsN (pediatric specialist, neonatologist), RN (medical doctor), BT (clinical research specialist), and PH (biostatistician).

Therapeutic levels of caffeine range between 5 and 25 mg/L (or μg/mL); the toxic level is 50 mg/L or higher [41]. During standard dosing regimens, concentrations are between 5–20 mg/L or 8–20 mg/L [42]; however, a recent study has shown that even with standard dosing, 15% of preterm infants have drug levels exceeding the considered toxic levels, without apparent adverse effects [43].

In our study, we will administer a higher dose of caffeine citrate than standard dosing regimens once during the hospital stay, without monitoring drug levels. According to the National Institute for Health and Care Excellence [44] (NICE guideline), caffeine citrate levels should be checked if preterm infants receive more than 20 mg/kg of caffeine citrate daily.

## Strength and limitation

This study, utilizing a randomized, controlled, open-label, two-armed design, will provide type A evidence on the effect of an additional loading dose of caffeine citrate on extubation success rate. Patients will be continuously monitored during hospitalization with data managed by an Independent Data Management Board.

As a limitation, we should mention that due to the open-label design, we cannot exclude performance bias. Furthermore, due to the strict inclusion criteria, the estimated sample size of 226 participants may extend the study period by a few years.

## Discussion

Here, we report an open-label, two-armed, randomized clinical trial invetigating the effect of the pre-extubational loading dose of caffeine citrate. Our main hypothesis is that this additional dose of caffeine citrate and elevated plasma levels in this rather vulnerable period, the peri-extubational period, will improve extubational success rate.

Reintubation has been shown to increase the risk of death, ventilation-associated pneumonia, pneumothorax, and neurological damage [14]. Different studies have already demonstrated the beneficial effect of higher doses of caffeine on extubational success rates [25, 27]; however, the use of daily higher doses of caffeine citrate has been investigated in several randomized trials for further beneficial effects, and the results are somewhat inconclusive [30]. In contrast, our study investigates the beneficial effect of a single higher dose at an appropriate time. Obtaining evidence for this feature may contribute to finding the optimal dosing regimen.

### Ethical approval and dissemination

The trial has been registered at ClinicalTrial.gov (NCT06401083) and received relevant ethical approval with the reference number OGYÉI/6838-11/2023 from the Hungarian Ethics Committee for Clinical Pharmacology of the Medical Research Council and National Institute of Pharmacy and Nutrition. At the end of the project, we will disseminate our results to the medical community. Authorship will be granted according to the criteria established by the International Committee of Medical Journal Editors. Results will be reported according to CONSORT guidelines [45].

### Trial status

The Hungarian Ethics Committee for Clinical Pharmacology of the Medical Research Council and the National Institute of Pharmacy and Nutrition has approved the study procedures. The protocol has been revised two times. Recruitment of preterm infants began in December 2023. The trial is planned to be completed in 2027. The first version of this protocol was completed on December 14, 2023.

## Supporting information

**S1 File. SPIRIT checklist.**
(DOCX)

**S2 File. Parental information and informed consent form.**
(PDF)

**S3 File. Summary of product characteristics (CITRATE DE CAFEINE COOPER 25 mg/ mL, injectable and oral solution).**
(PDF)

**S4 File. Case report form A.**
(PDF)

**S5 File. Case report form B.**
(PDF)

**S6 File. Case report form C.**
(PDF)

**S7 File. Case report form D.**
(PDF)

**S8 File. Graphical abstract.** Republished from Graphical abstract for NEOKOFF randomized clinical trial under a CC BY license, with permission from Fatina Hanna JD, original copyright 2024.
(TIF)

**S9 File. Study protocol.**
(DOCX)

## Acknowledgments

We would like to express our appreciation and gratitude to the nurses and clinicians actively participating in the enrollment and implementation of the study. The authors would also like to thank Andrea Valek and Miklós Szabó for providing access to the Hungarian National

Neonatal Database and Fatina Hanna for her great contribution to the preparation of the graphical abstract.

## Author Contributions

**Conceptualization:** Kinga Kovács, Rita Nagy, Lilla Andréka, Brigitta Teutsch, Miklós Szabó, Péter Varga, Péter Hegyi, Péter Hársfalvi, Ágnes Harmath, Csaba Nádor, Ákos Gasparics.

**Data curation:** Péter Hársfalvi, Ákos Gasparics.

**Formal analysis:** Ákos Gasparics.

**Funding acquisition:** Miklós Szabó, Péter Hegyi, Nándor Ács, Ágnes Harmath, Ákos Gasparics.

**Investigation:** Kinga Kovács, Rita Nagy, Ákos Gasparics.

**Methodology:** Kinga Kovács, Rita Nagy, Lilla Andréka, Brigitta Teutsch, Miklós Szabó, Péter Varga, Péter Hegyi, Ágnes Harmath, Csaba Nádor, Ákos Gasparics.

**Project administration:** Kinga Kovács, Lilla Andréka, Brigitta Teutsch, Miklós Szabó, Péter Varga, Nándor Ács, Ákos Gasparics.

**Resources:** Kinga Kovács, Brigitta Teutsch, Miklós Szabó, Péter Hegyi, Nándor Ács, Ágnes Harmath, Csaba Nádor, Ákos Gasparics.

**Software:** Kinga Kovács, Lilla Andréka, Brigitta Teutsch, Péter Hársfalvi, Ákos Gasparics.

**Supervision:** Kinga Kovács, Rita Nagy, Brigitta Teutsch, Miklós Szabó, Péter Varga, Péter Hegyi, Péter Hársfalvi, Nándor Ács, Ágnes Harmath, Csaba Nádor, Ákos Gasparics.

**Validation:** Kinga Kovács, Péter Hársfalvi, Ákos Gasparics.

**Visualization:** Kinga Kovács, Lilla Andréka, Péter Hársfalvi, Ákos Gasparics.

**Writing – original draft:** Kinga Kovács, Rita Nagy, Lilla Andréka, Brigitta Teutsch, Miklós Szabó, Péter Varga, Péter Hársfalvi, Ákos Gasparics.

**Writing – review & editing:** Kinga Kovács, Rita Nagy, Lilla Andréka, Brigitta Teutsch, Miklós Szabó, Péter Hegyi, Ágnes Harmath, Csaba Nádor, Ákos Gasparics.

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
