## [Decision Letter · Decision Letter 0]

25 Oct 2024

PONE-D-24-29935The effect of an additional pre-extubational loading dose of caffeine citrate on mechanically ventilated preterm infants (NEOKOFF trial): study protocol for a multicenter randomized clinical trialPLOS ONE

Dear Dr. Gasparics,

Thank you for submitting your manuscript to PLOS ONE. After careful consideration, we feel that it has merit but does not fully meet PLOS ONE’s publication criteria as it currently stands. Therefore, we invite you to submit a revised version of the manuscript that addresses the points raised during the review process.

We look forward to receiving your revised manuscript.

Kind regards,

Hasan Tolga Celik, M.D.

Academic Editor

PLOS ONE

Journal Requirements:

1. When submitting your revision, we need you to address these additional requirements. Please ensure that your manuscript meets PLOS ONE's style requirements, including those for file naming. The PLOS ONE style templates can be found at https://journals.plos.org/plosone/s/file?id=wjVg/PLOSOne_formatting_sample_main_body.pdf and https://journals.plos.org/plosone/s/file?id=ba62/PLOSOne_formatting_sample_title_authors_affiliations.pdf 2. We note that the original protocol that you have uploaded as other file contains an institutional logo. As this logo is likely copyrighted, we ask that you please remove it from this file and upload an updated version upon resubmission. 3. If any supporting files for review show as item type ‘other’ please change to item type ‘supporting info’ as the reviewer does not have access to these ’other’ files. 4. Thank you for stating the following financial disclosure: "The study will be financially sponsored by the Department of Obstetrics and Gynecology, Semmelweis University, Budapest.  Additional funding for the study has been provided by the grant TINL 3.2.1-21-2022-00011." Please state what role the funders took in the study.  If the funders had no role, please state: "The funders had no role in study design, data collection and analysis, decision to publish, or preparation of the manuscript." If this statement is not correct you must amend it as needed. Please include this amended Role of Funder statement in your cover letter; we will change the online submission form on your behalf. 5. When completing the data availability statement of the submission form, you indicated that you will make your data available on acceptance. We strongly recommend all authors decide on a data sharing plan before acceptance, as the process can be lengthy and hold up publication timelines. Please note that, though access restrictions are acceptable now, your entire data will need to be made freely accessible if your manuscript is accepted for publication. This policy applies to all data except where public deposition would breach compliance with the protocol approved by your research ethics board. If you are unable to adhere to our open data policy, please kindly revise your statement to explain your reasoning and we will seek the editor's input on an exemption. Please be assured that, once you have provided your new statement, the assessment of your exemption will not hold up the peer review process. 6. Your ethics statement should only appear in the Methods section of your manuscript. If your ethics statement is written in any section besides the Methods, please move it to the Methods section and delete it from any other section. Please ensure that your ethics statement is included in your manuscript, as the ethics statement entered into the online submission form will not be published alongside your manuscript.

Reviewers' comments:

Reviewer's Responses to Questions

**Comments to the Author**

1. Does the manuscript provide a valid rationale for the proposed study, with clearly identified and justified research questions?

Reviewer #1: Yes

Reviewer #2: Yes

Reviewer #3: Yes

2. Is the protocol technically sound and planned in a manner that will lead to a meaningful outcome and allow testing the stated hypotheses?

Reviewer #1: Yes

Reviewer #2: Yes

Reviewer #3: Yes

3. Is the methodology feasible and described in sufficient detail to allow the work to be replicable?

Reviewer #1: No

Reviewer #2: Yes

Reviewer #3: Yes

4. Have the authors described where all data underlying the findings will be made available when the study is complete?

Reviewer #1: Yes

Reviewer #2: Yes

Reviewer #3: Yes

5. Is the manuscript presented in an intelligible fashion and written in standard English?

Reviewer #1: Yes

Reviewer #2: Yes

Reviewer #3: Yes

6. Review Comments to the Author

You may also provide optional suggestions and comments to authors that they might find helpful in planning their study.

Reviewer #1: Dear Author,

It has been a real challenge to determine the guidelines for caffeine therapy in preterm infants. Therefore, I would like to thank you for your contribution, which I find very important. However, I would kindly request that you address these points below:

- Please clarify the extubation criteria that are used in the study.

- Please state the criteria for extubation failure used in the study that further leads to reintubation.

- Would the chronological age on the day of extubation be important? Could you please comment on this matter?

Yours sincerely.

Reviewer #2: Important note: This review pertains only to ‘statistical aspects’ of the study and so ‘clinical aspects’ [like medical importance, relevance of the study, ‘clinical significance and implication(s)’ of the whole study, etc.] are to be evaluated [should be assessed] separately/independently. Further please note that any ‘statistical review’ is generally done under the assumption that study specific methodological [as well as execution] issues are perfectly taken care of by the investigator(s). This review is not an exception to that and so does not cover clinical aspects {however, seldom comments are made only if those issues are intimately / scientifically related & intermingle with ‘statistical aspects’ of the study}. Agreed that ‘statistical methods’ are used as just tools here, however, they are vital part of methodology [and so should be given due importance]. I look at the manuscript in/with statistical view point, other reviewer(s) look(s) at it with different angle so that in totality the review is very comprehensive. However, there should be efforts from authors side to improve (may be by taking clues from reviewer’s comments). Therefore, please do not limit the revision only (with respect) to comments made here.

COMMENTS: Although there is no major flaw in the study/manuscript [and it is on very vital/important topic {namely “Extubation which involves removal of an endotracheal tube (ETT) from the trachea. Early and successful extubation can reduce complications such as ventilation acquired pneumonia (VAP)}], I have different opinion / observations/concerns or rather questions regarding a few issues which are given below:

Please refer to section on ‘Sample size’. As stated in line 119, “The calculated sample size is 226 patients (113 participants/arm)” is perfectly right, however, you need to adjust (inflate by 10% at least) the required sample size in the light of the fact that the proposed Chi-square test is ‘non-parametric’. Instead, one can/should apply continuity-corrected Z test and then the required sample size is Total 248 (124 in A, 124 in B). Moreover, note the Chi-square test is always two tailed [alternative hypothesis is two sided], whereas, you are interested in one sided alternative hypothesis (which may not matter much in this case, however, one should keep this thermotical point in mind. See the screen-shot below [from software ‘COMPRE2’ module of WIN-PEPI].

In line 202 you stated ‘’ drop-out rate is expected to be minimal due to the one-time intervention” seems to be incorrect [of doubtful value] as one-time intervention could be one of reasons of keeping the drop-out rate to the minimal level [it is not due]. Your claim perhaps sound diffently. As stated in lines 212-214 (Continuous variables will be compared using the t-test or Mann-Whitney test, whereas the χ2 test or Fisher’s exact test will be used for dichotomous outcomes.) following notes are pasted [in this context] from one famous standard textbook on ‘Medical Research Methodology’ {though I am sure that the authors already know these things}.

1. Though few variables are continuous in appearance they are likely to yield data that are in [at the most] ‘ordinal’ level of measurement [and not in ratio level of measurement for sure]. Then application of suitable non-parametric (or distribution free) test(s) is/are indicated/advisable then [even if distribution may be ‘Gaussian’ (also called ‘normal’)]. Agreed that there is/are no non-parametric test(s)/technique(s) available to be used as alternative in all situation(s), but should be used whenever/wherever they are available. Therefore, in short use suitable non-parametric test(s)/technique(s) while dealing with data that are in ‘ordinal’ level of measurement even if [despite that] the distribution may be ‘Gaussian’.

2.

the Fisher’s exact test’ [which is commonly available in many software] is applicable to 2x2 tables only [for larger tables Exact Chi-square test is available – reference ‘A network algorithm for performing Fisher's exact test in r× c contingency tables’ Cyrus R Mehta, Nitin R Patel, Journal of the American Statistical Association, 1983, Volume: 78, Issue: 382, Pages 427-434].

Kindly provide more convincing explanation (give details) of the statement in lines 224-5: Due to the nature of the study population, participants do not need to be blinded and neither those performing the intervention nor treating clinicians will be blinded. Why you say that “participants do not need to be blinded” is not clear. You may be [most likely be] 100% correct (considering age & status of the subjects). What about ‘equipoise’ principle? Kindly remember [please excuse me for such a harsh comment/statement] that this is a scientific/academic document and so all details should be clearly/correctly communicated (do not take reader’s for granted).

It is appreciable that in line 47-8 you stated “Preterm infants in the control group will receive the standard dosing regimen” and later lines 142-3 that “The control group will receive the above-mentioned routine dosing regimens” as it is very important that the control group should receive the routine (present standard of care) dosing regimens [/ active controls] in such trials. I suggest to mention this fact [handling of the control group] more prominently (as these are the only two occasions).

Good that Supporting information includes [S1 File:] SPIRIT checklist. But referring to CONSORT guidelines is also desirable. It is well-known that while reporting {findings from and even planning} ‘Clinical Trial’ one should follow CONSORT guidelines. Since your article type is protocol of ‘Clinical Trial’, you are supposed to cover these items in the report or even in ‘Protocol’ (even if you may not use them)]. Surprisingly, any information is not proposed to be displayed in tables. All important information is planned to be included only as ‘Supporting information’ and not in the main text [highly desirable].

Limitations (if any) of the study are not mentioned/listed anywhere. Does that mean {according to authors} there are none? As pointed out in ‘important note’ above “This review pertains only to ‘statistical aspects’ of the study and so ‘clinical aspects’ should be assessed separately/independently. In my opinion, to make this article more (completely) acceptable (which is quite possible and easy), a small amount of re-vision (re-drafting) may be needed. ‘Minor revision’ is recommended.

Reviewer #3: I had the pleasure to review the article titled “The effect of an additional pre-extubational loading dose of caffeine citrate on mechanically ventilated preterm infants (NEOKOFF trial): study protocol for a multicenter randomized clinical trials”.

The authors aim to investigate whether administering an additional loading dose of caffeine citrate one hour prior to extubation improves the success of extubation. The article is presented as a multicenter randomized clinical trial protocol.

Abstract:

- Aim: The specific aim or objective of the study should be explicitly stated in the abstract to clarify the central research question.

Methods:

- Inclusion Criteria: It would be helpful to specify whether there is a threshold for chronological age among the participants.

- Randomization: The randomization procedure mentions an allocation ratio of 1:1, but it would be beneficial to provide more clarity on how this will be achieved or operationalized within the trial.

7. PLOS authors have the option to publish the peer review history of their article (what does this mean?). If published, this will include your full peer review and any attached files.

Reviewer #1: **Yes: **Alper Aykanat

Reviewer #2: No

Reviewer #3: No

---

## [Author Response · Author response to Decision Letter 0]

14 Nov 2024

Dear Editors,

Thank you for allowing us to submit a revised draft of the manuscript (PONE-D-24-29935) “The effect of an additional pre-extubational loading dose of caffeine citrate on mechanically ventilated preterm infants (NEOKOFF trial): study protocol for a multicenter randomized clinical trial.”

We appreciate the time and effort you and the reviewers dedicated to providing feedback on our manuscript and are grateful for the insightful comments and valuable improvements to our paper. We have incorporated most of the suggestions made by the reviewers. 

Please see below a point-by-point response to the reviewers’ comments and concerns. All page numbers refer to the revised manuscript file with tracked changes.

Reviewers' Comments to the Authors:

RESPONSE TO REVIEWER #1: 

POINT 1: Please clarify the extubation criteria that are used in the study.

RESPONSE: Thank you very much for your comment. The literature indicates that while there are various protocols for assessing extubation readiness and criteria, uniform conclusions are lacking, similar to the case with caffeine. The discretion of the treating physician remains an important factor. See lines 88-95 and 192-193. Naturally, differences in extubation thresholds will exist between neonatal intensive care units (1-3).

However, we hypothesize that if the pre-extubation loading dose of caffeine citrate will improve the extubation success rate, it will be regardless of these differences. 

POINT 2: Please state the criteria for extubation failure used in the study that further leads to

reintubation.

RESPONSE: I would like to address the comment above; the manuscript has also been completed.

ACTION: Lines 239-241.

“As far as we know, there are still no standardized, evidence-based guidelines for intubation or reintubation (4), so reintubation will be performed based on the judgment of a senior clinician.”

POINT 3: Would the chronological age on the day of extubation be important? Could you please comment on this matter?

RESPONSE: Thank you for your comment. This would be quite an interesting observation. We cannot be certain, but it is presumed that caffeine treatment may be less effective (even more so at the same dosing regimens) in older individuals. This will become clear through secondary analyses. We will collect data regarding this information see S5 File: Case report form B.

RESPONSE TO REVIEWER #2: 

POINT 1: Please refer to section on ‘Sample size’. As stated in line 119, “The calculated sample size is 226 patients (113 participants/arm)” is perfectly right, however, you need to adjust (inflate by 10% at least) the required sample size in the light of the fact that the proposed Chi-square test is ‘non-parametric’. Instead, one can/should apply continuity-corrected Z test and then the required sample size is Total 248 (124 in A, 124 in B). Moreover, note the Chi- square test is always two tailed [alternative hypothesis is two sided], whereas, you are interested in one sided alternative hypothesis (which may not matter much in this case, however, one should keep this thermotical point in mind. See the screen-shot below [from software ‘COMPRE2’ module of WIN- PEPI].

RESPONSE: Thank you for the comment. The potential use of the Yates continuity correction was carefully considered during the design of the study. Due to the explorative nature of the investigation and the vulnerability of the specific study population, we decided to use the sample size calculation without the correction, as this results in a significantly more feasible enrollment with easier justification for the sample size from an ethical perspective. We decided to accept this trade-off despite it having the potential to slightly increase the type I error probability. This reasoning was accepted by the competent authority, and we received the ethical approval. 

POINT 2: In line 202 you stated ‘’ drop-out rate is expected to be minimal due to the one-time intervention” seems to be incorrect [of doubtful value] as one-time intervention could be one of reasons of keeping the drop-out rate to the minimal level [it is not due]. Your claim perhaps sound diffently.

RESPONSE: We expect a low drop-out rate due to the single intervention and short follow-up period. Since these preterm infants will already stay in the hospital, there is no risk of them not returning for follow-up. The question remains about the rate of late follow-up, but this would be considered a loss to follow-up rather than drop-out.

ACTION: See lines: 203-205

 “The drop-out rate is expected to be minimal due to the specific population and the short timeframe between the enrollment, randomization, and the one-time intervention.”

POINT 3: As stated in lines 212-214 (Continuous variables will be compared using the t-test or Mann-Whitney test, whereas the χ2 test or Fisher’s exact test will be used for dichotomous outcomes.) following notes are pasted [in this context] from one famous standard textbook on ‘Medical Research Methodology’ {though I am sure that the authors already know these things}.

1. Though few variables are continuous in appearance they are likely to yield data that are in [at the most] ‘ordinal’ level of measurement [and not in ratio level of measurement for sure]. Then application of suitable non-parametric (or distribution free) test(s) is/are indicated/advisable then [even if distribution may be ‘Gaussian’ (also called ‘normal’)]. Agreed that there is/are no non-parametric test(s)/technique(s) available to be used as alternative in all situation(s), but should be used whenever/wherever they are available. Therefore, in short use suitable non-parametric test(s)/technique(s) while dealing with data that are in ‘ordinal’ level of measurement even if [despite that] the distribution may be ‘Gaussian’.

2. the Fisher’s exact test’ [which is commonly available in many software] is applicable to 2x2 tables only [for larger tables Exact Chi-square test is available – reference ‘A network algorithm for performing Fisher's exact test in r× c contingency tables’ Cyrus R Mehta, Nitin R Patel, Journal of the American Statistical Association, 1983, Volume: 78, Issue: 382, Pages 427-434].

RESPONSE: Thank you very much for your comment, we extended the references based on your suggestion

ACTION: We inserted it as a reference. See line 217.

POINT 4: Kindly provide more convincing explanation (give details) of the statement in lines 224-5: Due to the nature of the study population, participants do not need to be blinded and neither those performing the intervention nor treating clinicians will be blinded. Why you say that “participants do not need to be blinded” is not clear. You may be [most likely be] 100% correct (considering age & status of the subjects). What about ‘equipoise’ principle? Kindly remember [please excuse me for such a harsh comment/statement] that this is a scientific/academic document and so all details should be clearly/correctly communicated (do not take reader’s for granted).

RESPONSE: Thank you very much for your comment. We clarified our claim regarding the study population in the manuscript. As far as the equipoise principle is concerned, we have addressed the uncertainties regarding caffeine administration in the manuscript (Lines: 112-122). We have highlighted that there is currently no high level of evidence supporting the definitive risks or benefits of caffeine use in this context. This genuine uncertainty supports the ethical foundation of our study and justifies the need for further research to establish clear evidence-based guidelines.

ACTION: See lines 227-229.

“Due to the nature of the study population (age and status of the neonates), participants do not need to be blinded and neither those performing the intervention nor treating clinicians will be blinded.”

POINT 5: It is appreciable that in line 47-8 you stated “Preterm infants in the control group will receive the standard dosing regimen” and later lines 142-3 that “The control group will receive the above-mentioned routine dosing regimens” as it is very important that the control group should receive the routine (present standard of care) dosing regimens [/ active controls] in such trials. I suggest to mention this fact [handling of the control group] more prominently (as these are the only two occasions).

RESPONSE: Thank you very much for your comment. We mentioned the handling of the control group throughout the manuscript. 

ACTION: See lines 187 and 237-239

„Arm “B” (control group) will receive a 20 mg/kg loading dose of caffeine citrate IV on the first day of life, then 5-10 mg/kg maintenance dose IV each day, also on the day of the extubation”

“Furthermore, caffeine half-life ranges from 20 to 100 hours, so during this period we expect significantly higher caffeine levels in the intervention group with the additional loading dose of caffeine-citrate compared to control group with standard care. “

POINT 6: Good that Supporting information includes [S1 File:] SPIRIT checklist. But referring to CONSORT guidelines is also desirable. It is well-known that while reporting {findings from and even planning} ‘Clinical Trial’ one should follow CONSORT guidelines. Since your article type is protocol of ‘Clinical Trial’, you are supposed to cover these items in the report or even in ‘Protocol’ (even if you may not use them)]. Surprisingly, any information is not proposed to be displayed in tables. All important information is planned to be included only as ‘Supporting information’ and not in the main text [highly desirable].

RESPONSE: Thank you very much for your comment. Meanwhile, CONSORT provides guidance for reporting the results; SPIRIT guidelines pertain to the protocol, so we have included the SPIRIT guidelines as a supplementary file. However, the results will be reported according to the CONSORT guidelines. We have declared it in the manuscript. 

ACTION: See lines:322-323.

“Results will be reported according to CONSORT guidelines.”

POINT 7: Limitations (if any) of the study are not mentioned/listed anywhere. Does that mean {according to authors} there are none? 

RESPONSE: Thank you very much for your comment. 

ACTION: Lines 296-303

“Strength and limitations”

This study, utilizing a randomized, controlled, open-label, two-armed design, will provide type A evidence on the effect of an additional loading dose of caffeine citrate on extubation success rate. Patients will be continuously monitored during hospitalization with data managed by an Independent Data Management Board. 

As a limitation, we should mention that due to the open-label design, we cannot exclude performance bias. Furthermore, due to the strict inclusion criteria, the estimated sample size of 226 participants may extend the study period by a few years.”

RESPONSE TO REVIEWER #3:

POINT 1: Aim: The specific aim or objective of the study should be explicitly stated in the abstract to clarify the central research question.

RESPONSE: Thank you very much for your comment. 

ACTION: We have made revisions accordingly. See lines 37-39.

„ Therefore, we aim to determine whether using a single additional loading dose of caffeine citrate one hour prior to extubation impacts the success rate of extubation.”

POINT 2: Methods: Inclusion Criteria: It would be helpful to specify whether there is a threshold for

chronological age among the participants.

RESPONSE: Thank you very much for your comment. There is no threshold regarding chronological age; however, as we mentioned, we include preterm neonates being treated at the neonatal intensive care units. We are aware that this somewhat defines chronological age, but there can be significant differences. As we responded to reviewer #1 POINT 3 we cannot be certain, but it is presumed that caffeine treatment may be less effective (even more so at the same dosing regimens) in older individuals. This will become clear through secondary analyses. We will collect data regarding this information see S5 File: Case report form B.

ACTION: We have made revisions accordingly. See line 150.

„The target population will be preterm infants treated in a tertiary intensive care unit at Semmelweis University regardless of chronological age.”

POINT 3: Randomization: The randomization procedure mentions an allocation ratio of 1:1, but it would be beneficial to provide more clarity on how this will be achieved or operationalized within the trial.

RESPONSE: Thank you for the comment. The allocation ratio, stratification, the type of randomization procedure (big-stick design), and both the platforms were included (the one on which the randomization list will be created and the one on which the actual randomization will be carried out). See lines: 221-225. Please clarify if additional information should be provided about the randomization procedure.

REFERENCE LIST:

1. Al-Mandari H, Shalish W, Dempsey E, Keszler M, Davis PG, Sant'Anna G. International survey on periextubation practices in extremely preterm infants. Arch Dis Child Fetal Neonatal Ed. 2015;100(5):F428-31.

2. Beltempo M, Isayama T, Vento M, Lui K, Kusuda S, Lehtonen L, et al. Respiratory Management of Extremely Preterm Infants: An International Survey. Neonatology. 2018;114(1):28-36.

3. Cheng Z, Dong Z, Zhao Q, Zhang J, Han S, Gong J, et al. A Prediction Model of Extubation Failure Risk in Preterm Infants. Frontiers in Pediatrics. 2021;9.

4. Tana M, Tirone C, Aurilia C, Lio A, Paladini A, Fattore S, et al. Respiratory Management of the Preterm Infant: Supporting Evidence-Based Practice at the Bedside. Children (Basel). 2023;10(3).

---

## [Decision Letter · Decision Letter 1]

2 Dec 2024

The effect of an additional pre-extubational loading dose of caffeine citrate on mechanically ventilated preterm infants (NEOKOFF trial): study protocol for a multicenter randomized clinical trial

PONE-D-24-29935R1

Dear Dr. Amos Gasparics,

We’re pleased to inform you that your manuscript has been judged scientifically suitable for publication and will be formally accepted for publication once it meets all outstanding technical requirements.

Kind regards,

Hasan Tolga Celik, M.D.

Academic Editor

PLOS ONE

Additional Editor Comments (optional):

Reviewers' comments:

Reviewer's Responses to Questions

**Comments to the Author**

1. Does the manuscript provide a valid rationale for the proposed study, with clearly identified and justified research questions?

Reviewer #1: Yes

Reviewer #2: Yes

Reviewer #3: Yes

2. Is the protocol technically sound and planned in a manner that will lead to a meaningful outcome and allow testing the stated hypotheses?

Reviewer #1: Yes

Reviewer #2: Partly

Reviewer #3: Yes

3. Is the methodology feasible and described in sufficient detail to allow the work to be replicable?

Reviewer #1: Yes

Reviewer #2: Yes

Reviewer #3: Yes

4. Have the authors described where all data underlying the findings will be made available when the study is complete?

Reviewer #1: Yes

Reviewer #2: Yes

Reviewer #3: Yes

5. Is the manuscript presented in an intelligible fashion and written in standard English?

Reviewer #1: Yes

Reviewer #2: Yes

Reviewer #3: Yes

6. Review Comments to the Author

You may also provide optional suggestions and comments to authors that they might find helpful in planning their study.

Reviewer #1: Thank you very much for your comments, which I find very comprehensive and adequate.

I am looking forward to see the final results of your study.

Reviewer #2: COMMENTS: All most all the comments made on earlier draft are/were considered positively & are attended {however not all reasons/arguments are very convincing}. Nevertheless, I recommend the acceptance.

Kindly note that statement [in one of the responses] “…given reasoning was accepted by the competent authority, and we received the ethical approval” is not sufficient/adequate (rather undesirable).

Reviewer #3: The authors have addressed all my comments thoroughly and to my satisfaction, demonstrating a clear effort to enhance the quality of the work.

7. PLOS authors have the option to publish the peer review history of their article (what does this mean?). If published, this will include your full peer review and any attached files.

Reviewer #1: **Yes: **Alper Aykanat

Reviewer #2: No

Reviewer #3: No

---

## [Editor Report · Acceptance letter]

31 Dec 2024

PONE-D-24-29935R1 

PLOS ONE

Dear Dr. Gasparics, 

I'm pleased to inform you that your manuscript has been deemed suitable for publication in PLOS ONE. Congratulations! Your manuscript is now being handed over to our production team.

Kind regards, 

on behalf of

Dr. Hasan Tolga Celik 

Academic Editor

PLOS ONE